# Learning differential equations that are easy to solve

**Jacob Kelly**[*]
University of Toronto, Vector Institute
jkelly@cs.toronto.edu

**Jesse Bettencourt**[*]
University of Toronto, Vector Institute
jessebett@cs.toronto.edu

**Matthew James Johnson**
Google Brain
mattjj@google.com

**David Duvenaud**
University of Toronto, Vector Institute
duvenaud@cs.toronto.edu

## Abstract

Differential equations parameterized by neural networks become expensive to solve numerically as training progresses. We propose a remedy that encourages learned dynamics to be easier to solve. Specifically, we introduce a differentiable surrogate for the time cost of standard numerical solvers, using higher-order derivatives of solution trajectories. These derivatives are efficient to compute with Taylor-mode automatic differentiation. Optimizing this additional objective trades model performance against the time cost of solving the learned dynamics. We demonstrate our approach by training substantially faster, while nearly as accurate, models in supervised classification, density estimation, and time-series modelling tasks.

## 1 Introduction

Differential equations describe a system's behavior by specifying its instantaneous dynamics. Historically, differential equations have been derived from theory, such as Newtonian mechanics, Maxwell's equations, or epidemiological models of infectious disease, with parameters inferred from observations. Solutions to these equations usually cannot be expressed in closed-form, requiring numerical approximation.

Recently, ordinary differential equations parameterized by millions of learned parameters, called neural ODEs, have been fit for latent time series models, density models, or as a replacement for very deep neural networks (Rubanova et al., 2019; Grathwohl et al., 2019; Chen et al., 2018). These learned models are not constrained to match a theoretical model, only to optimize an objective on observed data. Learned models with nearly indistinguishable predictions can have substantially different dynamics. This raises the possibility that we can find equivalent models that are easier and faster to solve. Yet standard training methods have no way to penalize the complexity of the dynamics being learned.

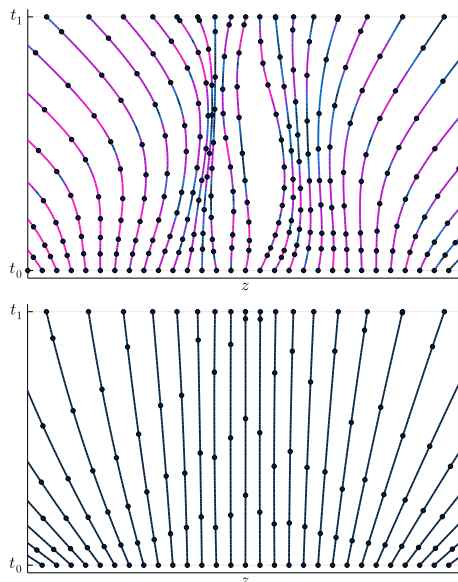

Figure 1: *Top:* Trajectories of an ODE fit to map $\mathbf{z}(t_1) = \mathbf{z}(t_0) + \mathbf{z}(t_0)^3$. The learned dynamics are unnecessarily complex and require many evaluations (black dots) to solve. *Bottom:* Regularizing the third total derivative $\frac{\mathrm{d}^3 \mathbf{z}(t)}{\mathrm{d}t^3}$ (shown by colour) gives dynamics that fit the same map, but require fewer evaluations to solve.

---

[*]Equal Contribution. Code available at:
github.com/jacobjinkelly/easy-neural-ode

How can we learn dynamics that are faster to solve numerically without substantially changing their predictions? Much of the computational advantages of a continuous-time formulation come from using adaptive solvers, and most of the time cost of these solvers comes from repeatedly evaluating the dynamics function, which in our settings is a moderately-sized neural network. So, we'd like to reduce the number of function evaluations (NFE) required for these solvers to reach a given error tolerance. Ideally, we would add a term penalizing the NFE to the training objective, and let a gradient-based optimizer trade off between solver cost and predictive performance. But because NFE is integer-valued, we need to find a differentiable surrogate.

The NFE taken by an adaptive solver depends on how far it can extrapolate the trajectory forward without introducing too much error. For example, for a standard adaptive-step Runge-Kutta solver with order $m$, the step size is approximately inversely proportional to the norm of the local $m$th total derivative of the solution trajectory with respect to time. That is, a larger $m$th derivative leads to a smaller step size and thus more function evaluations. Thus, we propose to minimize the norm of this total derivative during training, as a way to control the time required to solve the learned dynamics.

In this paper, we investigate the effect of this speed regularization in various models and solvers. We examine the relationship between the solver order and the regularization order, and characterize the tradeoff between speed and performance. In most instances, we find that solver speed can be approximately doubled without a substantial increase in training loss. We also provide an extension to the JAX program transformation framework that provides Taylor-mode automatic differentiation, which is asymptotically more efficient for computing the required total derivatives than standard nested gradients.

Our work compares against and generalizes that of Finlay et al. (2020), who proposed regularizing dynamics in the FFJORD density estimation model, and showed that it stabilized dynamics enough in that setting to allow the use of fixed-step solvers during training.

## 2 Background

An ordinary differential equation (ODE) specifies the instantaneous change of a vector-valued state $\mathbf{z}(t)$: $\frac{d\mathbf{z}(t)}{dt} = f(\mathbf{z}(t), t, \theta)$. Given an initial condition $\mathbf{z}(t_0)$, computing the state at a later time:

$$\mathbf{z}(t_1) = \mathbf{z}(t_0) + \int_{t_0}^{t_1} f(\mathbf{z}(t), t, \theta) \, dt$$

is called an initial value problem (IVP). For example, $f$ could describe the equations of motion for a particle, or the transmission and recovery rates for a virus across a population. Usually, the required integral has no analytic solution, and must be approximated numerically.

**Adaptive-step Runge-Kutta ODE Solvers**  Runge-Kutta methods (Runge, 1895; Kutta, 1901) approximate the solution trajectories of ODEs through a series of small steps, starting at time $t_0$. At each step, they choose a step size $h$, and fit a local approximation to the solution, $\hat{\mathbf{z}}(t)$, using several evaluations of $f$. When $h$ is sufficiently small, the numerical error of a $m$th-order method is bounded by $\|\hat{\mathbf{z}}(t + h) - \mathbf{z}(t + h)\| \le ch^{m+1}$ for some constant $c$ (Hairer et al., 1993). So, for a $m$th-order method, the local error grows approximately in proportion to the size of the $m$th coefficient in the Taylor expansion of the true solution. All else being equal, controlling this coefficient for all dimensions of $\mathbf{z}(t)$ will allow larger steps to be taken without surpassing the error tolerance.

**Neural Ordinary Differential Equations**  The dynamics function $f$ can be a moderately-sized neural network, and its parameters $\theta$ trained by gradient descent. Solving the resulting IVP is analogous to evaluating a very deep residual network in which the number of layers corresponds to the number of function evaluations of the solver (Chang et al., 2017; Ruthotto & Haber, 2018; Chen et al., 2018). Solving such continuous-depth models using adaptive numerical solvers has several computational advantages over standard discrete-depth network architectures. However, this approach is often slower than using a fixed-depth network, due to an inability to control the number of steps required by an adaptive-step solver.

## 3 Regularizing Higher-Order Derivatives for Speed

The ability of Runge-Kutta methods to take large and accurate steps is limited by the $K$th-order Taylor coefficients of the solution trajectory. We would like these coefficients to be small. Specifically, we propose to regularize the squared norm of the $K$th-order total derivatives of the state with respect to time, integrated along the entire solution trajectory:

$$\mathcal{R}_K(\theta) = \int_{t_0}^{t_1} \left\| \frac{\mathrm{d}^K \mathbf{z}(t)}{\mathrm{d}t^K} \right\|_2^2 \mathrm{d}t \tag{1}$$

where $\|\cdot\|_2^2$ is the squared $\ell_2$ norm, and the dependence on the dynamics parameters $\theta$ is implicit through the solution $\mathbf{z}(t)$ integrating $\frac{\mathrm{d}\mathbf{z}(t)}{\mathrm{d}t} = f(\mathbf{z}(t), t, \theta)$. During training, we weigh this regularization term by a hyperparameter $\lambda$ and add it to our original loss to get our regularized objective:

$$L_{reg}(\theta) = L(\theta) + \lambda \mathcal{R}_K(\theta) \tag{2}$$

What kind of solutions are allowed when $R_K = 0$? For $K = 0$, we have $\|\mathbf{z}(t)\|_2^2 = 0$, so the only possible solution is $\mathbf{z}(t) = 0$. For $K = 1$, we have $\|f(\mathbf{z}(t), t)\|_2^2 = 0$, so all solutions are constant, flat trajectories. For $K = 2$ solutions are straight-line trajectories. Higher values of $K$ shrink higher derivatives, but don't penalize lower-order dynamics. For instance, a quadratic trajectory will have $\mathcal{R}_3 = 0$. Setting the $K$th order dynamics to exactly zero everywhere automatically makes all higher orders zero as well. Figure 1 shows that regularizing $\mathcal{R}_3$ on a toy 1D neural ODE reduces NFE.

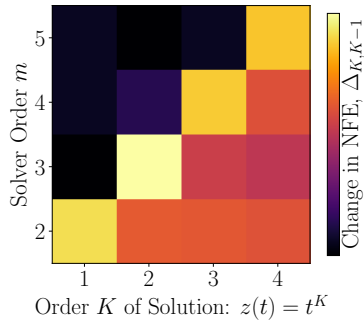

Which orders should we regularize? We propose matching the order of the regularizer to that of the solver being used. We conjecture that regularizing dynamics of lower orders than that of the solver restricts the model unnecessarily, and that letting the lower orders remain unregularized should not increase NFE very much. Figure 2 shows empirically which orders of Runge-Kutta solvers can efficiently solve which orders of toy polynomial trajectories. We partially confirm these conjectures on real models and datasets in section 6.2.

Figure 2: $m$-order Runge-Kutta solvers need small steps when the dynamics have non-zero total derivatives of order $K \geq m$ (lower triangle). Color denotes the increase in number of steps from $K$ to $K-1$, normalized for each solver order.

The solution trajectory and our regularization term can be computed in a single call to an ODE solver by augmenting the system with the integrand in eq. (1).

## 4 Efficient Higher Order Differentiation with Taylor Mode

The number of terms in higher-order forward derivatives grows exponentially in $K$, becoming prohibitively expensive for $K = 5$, and causing substantial slowdowns even for $K = 2$ and $K = 3$. Luckily, there exists a generalization of forward-mode automatic differentiation (AD), known as Taylor mode, which can compute the total derivative exactly for a cost of only $\mathcal{O}(K^2)$. We found that this asymptotic improvement reduced wall-clock time by an order of magnitude, even for $K$ as low as 3.

**First-order forward-mode AD**   Standard forward-mode AD computes, for a function $f(x)$ and an input perturbation vector $v$, the product $\frac{\partial f}{\partial x} v$. This Jacobian-vector product, or JVP, can be computed efficiently without explicitly instantiating the Jacobian. This implicit computation of JVPs is straightforward whenever $f$ is a composition of operations for which which implicit JVP rules are known.

**Higher-order Jacobian-vector products**   Forward-mode AD can be generalized to higher orders to compute $K$th-order Jacobians contracted $K$ times against the perturbation vector: $\frac{\partial^K f}{\partial x^K} v^{\otimes K}$. Similarly, this can also be computed without representing any Jacobian matrices explicitly.

A naïve approach to higher-order forward mode is to recursively apply first-order forward mode. Specifically, nesting JVPs $K$ times gives the right answer: $\frac{\partial^K f}{\partial x^K} v^{\otimes K} = \frac{\partial}{\partial x}(\cdots(\frac{\partial}{\partial x}(\frac{\partial f}{\partial x}v)v)\cdots v)$ but causes an unnecessary exponential slowdown, costing $O(\exp(K))$. This is because expressions that appear in lower derivatives also appear in higher derivatives, but the work to compute is not shared across orders.

**Taylor Mode**  Taylor-mode AD generalizes first-order forward mode to compute the first $K$ derivatives exactly with a time cost of only $O(K^2)$ or $O(K \log K)$, depending on the operations involved. Instead of providing rules for propagating perturbation vectors, one provides rules for propagating truncated Taylor series. Some example rules are shown in table 1. For more details see the Appendix and Griewank & Walther (2008, Chapter 13). We provide an open source implementation of Taylor mode AD in the JAX Python library (Bradbury et al., 2018).

| Function | Taylor propagation rule |
|---|---|
| $y = z + cw$ | $y_{[k]} = z_{[k]} + cw_{[k]}$ |
| $y = z * w$ | $y_{[k]} = \sum_{j=0}^{k} z_{[j]} w_{[k-j]}$ |
| $y = z/w$ | $y_{[k]} = \frac{1}{w_0}\left[ z_k - \sum_{j=0}^{k-1} z_{[j]} w_{[k-j]} \right]$ |
| $y = \exp(z)$ | $\tilde{y}_{[k]} = \sum_{j=1}^{k} y_{[k-j]} \tilde{z}_{[j]}$ |
| $s = \sin(z)$ | $\tilde{s}_{[k]} = \sum_{j=1}^{k} \tilde{z}_{[j]} c_{[k-j]}$ |
| $c = \cos(z)$ | $\tilde{c}_{[k]} = \sum_{j=1}^{k} -\tilde{z}_{[j]} s_{[k-j]}$ |

Table 1: Rules for propagating Taylor polynomial coefficients through standard functions. These rules generalize standard first-order derivatives. Notation $z_{[i]} = \frac{1}{i!} z_i$ and $\tilde{y}_{[i]} = \frac{i}{i!} z_i$.

## 5  Experiments

We consider three different tasks in which continuous-depth or continuous time models might have computational advantages over standard discrete-depth models: supervised learning, continuous generative modeling of time-series (Rubanova et al., 2019), and density estimation using continuous normalizing flows (Grathwohl et al., 2019). Unless specified otherwise, we use the standard `dopri5` Runge-Kutta 4(5) solver (Dormand & Prince, 1980; Shampine, 1986).

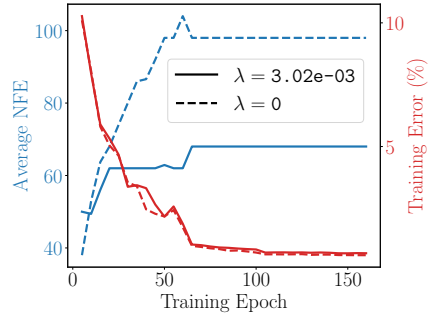

Figure 3: Number of function evaluations (NFE) and training error during training. Speed regularization (solid) decreases the NFE throughout training without substantially changing the training error.

### 5.1  Supervised Learning

We construct a model for MNIST classification: it takes in as input a flattened MNIST image and integrates it through dynamics given by a simple MLP, then applies a linear classification layer. In fig. 3 we compare the NFE and training error of a model with and without regularizing $\mathcal{R}_3$.

### 5.2  Continuous Generative Time Series Models

As in Rubanova et al. (2019), we use the Latent ODE architecture for modelling trajectories of ICU patients using the PhysioNet Challenge 2012 dataset (Silva et al., 2012). This variational autoencoder architecture uses an RNN recognition network, and models the state dynamics using an ODE in a latent space.

In the supervised learning setting described in the previous section only the final state affects model predictions. In contrast, time-series models' predictions also depend on the value of the trajectory at all intermediate times when observations were made. So, we might expect speed regularization to be ineffective due to these extra constraints on the dynamics. However, fig. 4 shows that, without changing their overall shape the latent dynamics can be adjusted to reduce their NFE by a factor of 3.

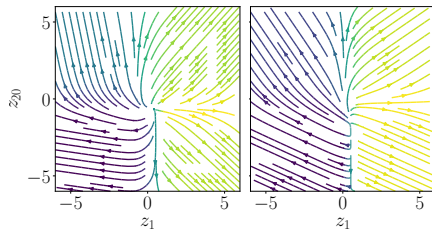

(a) Unregularized       (b) Regularized

Figure 4: Regularizing dynamics in a latent ODE modeling PhysioNet clinical data. Shown are a representative 2-dimensional slice of 20 dimensional dynamics. We reduce average NFE from 281 to 90 while only incurring an 8% increase in loss.

## 5.3 Density Estimation with Continuous Normalizing Flows

Our third task is unsupervised density estimation, using a scalable variant of continuous normalizing flows called FFJORD (Grathwohl et al., 2019). We fit the MINIBOONE tabular dataset from Papamakarios et al. (2017) and the MNIST image dataset (LeCun et al., 2010). We use the respective singe-flow architectures from Grathwohl et al. (2019).

Grathwohl et al. (2019) noted that the NFE required to numerically integrate their dynamics could become prohibitively expensive throughout training. Table 2 shows that we can reduce NFE by 38% for only a 0.6% increase in log-likelihood measured in bits/dim.

**How to train your Neural ODE**     We compare against the approach of Finlay et al. (2020), who design two regularization terms specifically for stabilizing the dynamics of FFJORD models:

$$\mathcal{K}(\theta) = \int_{t_0}^{t_1} \|f(\mathbf{z}(t), t, \theta)\|_2^2 \ \mathrm{d}t \tag{3}$$

$$\mathcal{B}(\theta) = \int_{t_0}^{t_1} \|\epsilon^\mathsf{T} \nabla_\mathbf{z} f(\mathbf{z}(t), t, \theta)\|_2^2 \ \mathrm{d}t, \qquad \epsilon \sim \mathcal{N}(0, I) \tag{4}$$

The first term is designed to encourage straight-line paths, and the second, stochastic, term is designed to reduce overfitting. Finlay et al. (2020) used fixed-step solvers during training for some datasets. We compare these two regularization on training with each of adaptive and fixed-step solvers, and evaluated using an adaptive solver, in section 6.3.

# 6   Analysis and Discussion

## 6.1   Trading off function evaluations for loss

What does the trade off between accuracy and speed look like? Ideally, we could reduce the solver time a lot without substantially reducing model performance. Indeed, this is demonstrated in all three settings we explored. Figure 5 shows that generally, model performance starts getting substantially worse only after a 50% reduction in solver speed when controlling $\mathcal{R}_2$.

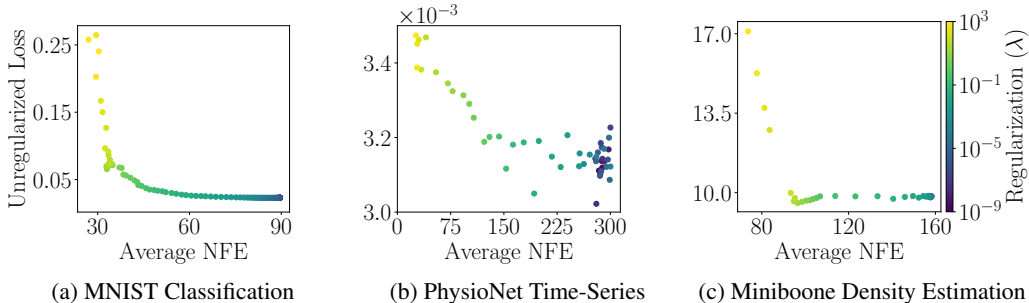

(a) MNIST Classification      (b) PhysioNet Time-Series      (c) Miniboone Density Estimation

Figure 5: Tuning the regularization of $\mathcal{R}_2$ trades off between training loss and solver speed in three different applications of neural ODEs. Horizontal axes show average number of function evaluations, and vertical axes show unregularized training loss, both at the end of training.

## 6.2   Order of regularization vs. order of solver

Which order of total derivatives should we regularize for a particular solver? As mentioned earlier, we conjecture that the best choice would be to match the order of the solver being used. Regularizing too low an order might needlessly constrain the dynamics and make it harder to fit the data, while regularizing too high an order might leave the dynamics difficult to solve for a lower-order solver. However, we also expect that optimizing higher-order derivatives might be challenging, since these higher derivatives can change quickly even for small changes to the dynamics parameters.

Figures 6 and 7 investigate this question on the task of MNIST classification. Figure 6 compares the effectiveness of regularizing different orders when using a solver of a particular order. For a 2nd order solver, regularizing $K = 2$ produces a strictly better trade-off between performance and speed, as expected. For higher-order solvers, including ones with adaptive order, we found that regularizing orders above $K = 3$ gave little benefit.

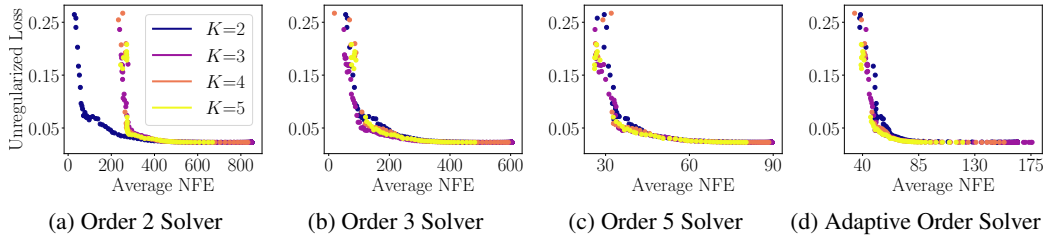

(a) Order 2 Solver     (b) Order 3 Solver     (c) Order 5 Solver     (d) Adaptive Order Solver

Figure 6: Comparing tradeoff between speed and performance when regularizing different orders.
6a): For a 2nd-order solver, regularizing the 2nd total derivative gives the best tradeoff. 6b): For
a 3rd-order solver, regularizing the 3rd total derivative gives the best tradeoff, but the difference is
small. 6c): For a 5th-order solver, results are mixed. 6d): For an adaptive-order solver, the difference
is again small but regularizing higher orders works slightly better.

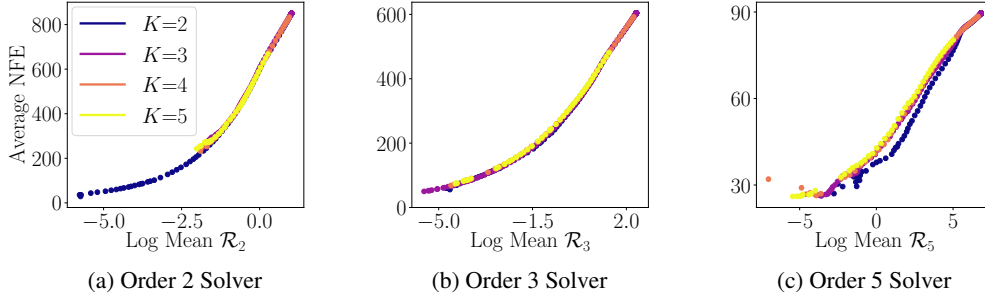

(a) Order 2 Solver       (b) Order 3 Solver       (c) Order 5 Solver

Figure 7: For all orders, $\mathcal{R}_K$ varies monotonically with NFE. For each order of solver, the model
with the lowest NFE was achieved by regularizing the same order.

Figure 7 investigates the relationship between $\mathcal{R}_K$ and the quantity it is meant to be a surrogate
for: NFE. We observe a clear monotonic relationship between the two, for all orders of solver and
regularization.

### 6.3 Do we reduce training time?

Our approach produces models that are fastest to evaluate at test time. However, when we train
with adaptive solvers we do not improve overall training time, due to the additional expense of
computing our regularizer. Training with a fixed-grid solver is faster, but can be unstable if dynamics
are unregularized. Finlay et al. (2020)'s regularization and ours allow us to use fixed grid solvers and
reduce training time. However, ours is $2.4\times$ slower than Finlay et al. (2020) for FFJORD because
their regularization re-uses terms already computed in the FFJORD training objective. For objectives
where these cannot be re-used, like MNIST classification, our method is $1.7\times$ slower, but achieves
better test-time NFE.

### 6.4 Are we making the solver overconfident?

Because we optimize dynamics in a way specifically designed to make the solver take longer steps,
we might fear that we are "adversarially attacking" our solver, making it overconfident in its ability
to extrapolate. Figure 8a shows that this is not the case for MNIST classification.

### 6.5 Does speed regularization overfit?

Finlay et al. (2020) motivated one of their regularization terms by the possibility of overfitting: having
faster dynamics only for the examples in the training set, but still low on the test set. However, they
did not check whether overfitting was occurring. In fig. 8b we confirm that our regularized dynamics
have nearly identical average solve time on a held-out test set, on MNIST classification.

## 7 Related Work

Grathwohl et al. (2019) mention attempting to use weight decay and spectral normalization to reduce
NFE. Of course, Finlay et al. (2020), among other contributions, regularized trajectories of continuous

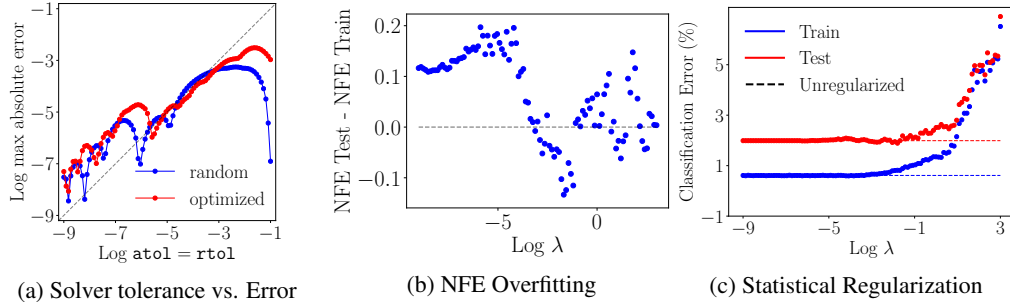

(a) Solver tolerance vs. Error     (b) NFE Overfitting     (c) Statistical Regularization

Figure 8: Figure 8a: We observe that the actual solver error is about equally well-calibrated for regularized dynamics as random dynamics, indicating that regularization does not make the solver overconfident. Figure 8b: There is negligible overfitting of solver speed. Figure 8c: Speed regularization does not usefully improve generalization. For large $\lambda$, our method reduces overfitting, but increases overall test error due to under-fitting.

normalizing flows and introduced the use of fixed-step solvers for stable and faster training. The use of fixed-step solves is also explored in Onken & Ruthotto (2020). Onken et al. (2020) also regularized the trajectories of continuous normalizing flows, among other contributions. Massaroli et al. (2020b) introduce new formulations of neural differential equations and investigate regularizing these models as applied to continuous normalizing flows in Massaroli et al. (2020a).

Poli et al. (2020) introduce solvers parameterized by neural networks for faster solving of neural differential equations. Kidger et al. (2020a) exploit the structure of the adjoint equations to construct a solver needing less NFE to speed up backpropgation through neural differential equations.

Morrill et al. (2020) introduce a method to improve the speed of neural controlled differential equations (Kidger et al., 2020b) which are designed for with irregularly-sampled timeseries.

Simard et al. (1991) regularized the dynamics of discrete-time recurrent neural networks to improve their stability, by constraining the norm of the Jacobian of the dynamics function in the direction of its largest eigenvalue. However, this approach has an $\mathcal{O}(D^3)$ time cost. De Brouwer et al. (2019) introduced a parameterization of neural ODEs analogous to instantaneous Gated Recurrent Unit (GRU) recurrent neural network architectures in order to stabilize training dynamics. Dupont et al. (2019) provided theoretical arguments that adding extra dimensions to the state of a neural ODE should make training easier, and showed that this helped reduce NFE during training.

Chang et al. (2017) noted the connection between residual networks and ODEs, and took advantage of this connection to gradually make resnets deeper during training, in order to save time. One can view the increase in NFE while neural ODEs as an automatic, but uncontrolled, version of their

Table 2: Density Estimation on MNIST using FFJORD. For adaptive solvers, indicated by $\infty$ Steps, our approach is slowest to train, but requires the fewest NFE once trained. For fixed-step solvers our approach achieves lower bits/dim and NFE when comparing across fixed-grid solvers using the same number of steps. Fixed step solvers that diverged due to instability are indicated by NaN bits/dim.

| | Training | | Evaluation using adaptive solvers | | | | |
| | Steps | Hours | Bits/Dim | NFE | $\mathcal{R}_2$ | $\mathcal{B}$ | $\mathcal{K}$ |
|---|---|---|---|---|---|---|---|
| Unregularized | 8 | - | NaN | - | - | - | - |
| | $\infty$ | 35.8 | **1.033** | 149 | 3596 | 4.76 | 73.6 |
| RNODE | 5 | - | NaN | - | - | - | - |
| (Finlay et al., 2020) | 6 | **8.4** | 1.069 | 122 | 157.8 | 1.82 | 35.0 |
| | 8 | 11.1 | 1.048 | 97 | 39.3 | 1.85 | 34.8 |
| | $\infty$ | 22.9 | 1.049 | 104 | 46.6 | 1.85 | 34.7 |
| TayNODE (ours) | 5 | 20.3 | 1.077 | 98 | 31.3 | 2.89 | 36.5 |
| | 6 | 20.4 | 1.057 | 105 | 31.1 | 2.91 | 36.5 |
| | 8 | 27.1 | 1.046 | 98 | 26.0 | 2.53 | 36.3 |
| | $\infty$ | 54.7 | 1.039 | **92** | 22.9 | 2.41 | 36.2 |

method. Their results suggest we might benefit from introducing a speed regularization schedule that gradually tapers off during training.

Novak et al. (2018); Drucker & LeCun (1992) regularized the gradients of neural networks to improve generalization.

We speculate on the application of our regularization in eq. (1) for other purposes, including adversarial robustness (Yang et al., 2020; Hanshu et al., 2019). and function approximation with Gaussian processes (Dutra et al., 2014; van der Vaart et al., 2008).

## 8 Scope

The initial speedups obtained in this paper are not yet enough to make neural ODEs competitive with standard fixed-depth architectures in terms of speed for standard supervised learning. However, there are many applications where continuous-depth architectures provide a unique advantage. Besides density models such as FFJORD and time series models, continuous-depth architectures have been applied in solving mean-field games (Ruthotto et al., 2019), image segmentation (Pinckaers & Litjens, 2019), image super-resolution (Scao, 2020), and molecular simulations (Wang et al., 2020). These applications, which already use continuous-time models, could benefit from the speed regularization proposed in this paper.

While we investigated only ODEs in this paper, this approach could presumably be extended straightforwardly to neural stochastic differential equations fit by adaptive solvers (Li et al., 2020) and other flavors of parametric differential equations fit by gradient descent (Rackauckas et al., 2019).

## 9 Limitations

**Hyperparameters**   The hyperparameter $\lambda$ needs to be chosen to balance speed and training loss. One the other hand, neural ODEs don't require choosing the outer number of layers, which needs to be chosen separately for each stack of layers in standard architectures.

One also needs to choose solver order and tolerances, and these can substantially affect solver speed. We did not investigate loosening tolerances, or modifying other parameters of the solver. The default tolerance of `1.4e-8` for both `atol` and `rtol` behaved well in all our experiments.

One also needs to choose $K$. Higher $K$ seems to generally work better, but is slower per step at training time. In principle, if one can express their utility explicitly in terms of training loss and NFE, it may be possible to tune $\lambda$ automatically during training using the predictable relationship between $\mathcal{R}_K$ and NFE shown in fig. 7.

**Slower overall training**   Although speed regularization reduces the overall NFE during training, it makes each step more expensive. In our density estimation experiments (table 2), the overall effect was about about 70% slower training, compared to no regularization, when using adaptive solvers. However, test-time evaluation is much faster, since there is no slowdown per step.

## 10 Conclusions

This paper is an initial attempt at controlling the integration time of differential equations by regularizing their dynamics. This is an almost unexplored problem, and there are almost certainly better quantities to optimize than the ones examined in this paper.

Based on these initial experiments, we propose three practical takeaways:

1. Across all tasks, tuning the regularization usually gave at least a 2x speedup without substantially hurting model performance.

2. Overall training time with speed regularization is in general about 30% to 50% slower with adaptive solvers.

3. For standard solvers, regularizing orders higher than $\mathcal{R}_2$ or $\mathcal{R}_3$ provided little additional benefit.

**Future work** It may be possible to adapt solver architectures to take advantage of flexibility in choosing the dynamics. Standard solver design has focused on robustly and accurately solving a given set of differential equations. However, in a learning setting, we could consider simply rejecting some kinds of dynamics as being too difficult to solve, analogous to other kinds of constraints we put on models to encourage statistical regularization.

## Acknowledgements

We thank Dougal Maclaurin, Andreas Griewank, Barak Perlmutter, Ken Jackson, Chris Finlay, James Saunderson, James Bradbury, Ricky T.Q. Chen, Will Grathwohl, Chris Rackauckas, David Sanders, and Lyndon White for feedback and helpful discussions. Resources used in preparing this research were provided, in part, by the Province of Ontario, the Government of Canada through CIFAR, NSERC, and companies sponsoring the Vector Institute.

## Broader Impact

We expect the main impact from this work, if any, would be through a potential improvement of the fundamental modeling tools of regression, classification, time series models, and density estimation. Thus the impact of this work is not distinct from that of improved machine learning tools in general. While machine learning tools present both benefits and unintended consequences, we avoid speculating further.

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
