[Supplementary Material]

## Appendix A Taylor-mode Automatic Differentiation

### A.1 Taylor Polynomials

To clarify the relationship between the presentation in Chapter 13 of Griewank & Walther (2008) and our results we give the distinction between the Taylor coefficients and derivative coefficients, also known, unhelpfully, as *Tensor* coefficients.

For a sufficiently smooth vector valued function $f : \mathbb{R}^n \to \mathbb{R}^m$ and the polynomial

$$x(t) = x_{[0]} + x_{[1]}t + x_{[2]}t^2 + x_{[3]}t^3 + \cdots + x_{[d]}t^d \in \mathbb{R}^n \tag{5}$$

we are interested in the $d$-truncated Taylor expansion

$$y(t) = f(x(t)) + O(t^{d+1}) \tag{6}$$
$$\equiv y_{[0]} + y_{[1]}t + y_{[2]}t^2 + y_{[3]}t^3 + \cdots + y_{[d]}t^d \in \mathbb{R}^m \tag{7}$$

with the notation that $y_{[i]} = \frac{1}{i!}y_i$ is the *Taylor coefficient*, which is the normalized *derivative coefficient* $y_i$.

The Taylor coefficients of the expansion, $y_{[j]}$, are smooth functions of the $i \leq j$ coefficients $x_{[i]}$,

$$y_{[0]} = y_{[0]}(x_{[0]}) \qquad\qquad = f(x_{[0]}) \tag{8}$$
$$y_{[1]} = y_{[1]}(x_{[0]}, x_{[1]}) \qquad\qquad = f'(x_{[0]})x_{[1]} \tag{9}$$
$$y_{[2]} = y_{[2]}(x_{[0]}, x_{[1]}, x_{[2]}) \qquad\qquad = f'(x_{[0]})x_{[2]} + \frac{1}{2}f''(x_{[0]})x_{[1]}x_{[1]} \tag{10}$$
$$y_{[3]} = y_{[3]}(x_{[0]}, x_{[1]}, x_{[2]}, x_{[3]}) \qquad = f'(x_{[0]})x_{[3]} + f''(x_{[0]})x_{[1]}x_{[2]} + \frac{1}{6}f'''(x_{[0]})x_{[1]}x_{[1]}x_{[1]} \tag{11}$$
$$\vdots$$

These, as given in Griewank & Walther (2008), are written in terms of the normalized, Taylor coefficients. This obscures their direct relationship with the derivatives, which we make explicit.

Consider the polynomial eq. (5) with Taylor coefficients expanded so their normalization is clear. Further, let's use suggestive notation that these coefficients correspond to the higher derivatives of of $x$ with respect to $t$, making $x(t)$ a Taylor polynomial. That is $x_{[i]} = \frac{1}{i!}x_i = \frac{1}{i!}\frac{d^i x}{dt^i}$.

$$x(t) = x_0 + x_1 t + \frac{1}{2!}x_2 t^2 + \frac{1}{3!}x_3 t^3 + \cdots + \frac{1}{d!}x_d t^d \in \mathbb{R}^n \tag{12}$$
$$= x_0 + \frac{dx}{dt}t + \frac{1}{2!}\frac{d^2 x}{dt^2}t^2 + \frac{1}{3!}\frac{d^3 x}{dt^3}t^3 + \cdots + \frac{1}{d!}\frac{d^d x}{dt^d}t^d \in \mathbb{R}^n \tag{13}$$
$$\tag{14}$$

Again, we are interested in the polynomial eq. (7), but with the normalization terms explicit

$$y(t) \equiv y_0 + y_1 t + \frac{1}{2!}y_2 t^2 + \frac{1}{3!}y_3 t^3 + \cdots + \frac{1}{d!}y_d t^d \in \mathbb{R}^m \tag{15}$$

Now we can expand the expressions for the Taylor coefficients $y_{[i]}$ to expressions for derivative coefficients $y_i = i!y_{[i]}$

The coefficients of the Taylor expansion, $y_j$, are smooth functions of the $i \leq j$ coefficients $x_i$,

$$
\begin{aligned}
y_0 = y_0(x_0) \qquad &= y_{[0]}(x_0) \\
&= f(x_0) \qquad\qquad\qquad\qquad\qquad\qquad (16)
\end{aligned}
$$

$$
\begin{aligned}
y_1 = y_1(x_0, x_1) \qquad &= y_{[1]}(x_0, x_1) \\
&= f'(x_0) x_1 \\
&= f'(x_0)\frac{\mathrm{d}x}{\mathrm{d}t} \qquad\qquad\qquad\qquad\quad (17)
\end{aligned}
$$

$$
\begin{aligned}
y_2 = y_2(x_0, x_1, x_2) \qquad &= 2!\left( y_{[2]}\left(x_0, x_1, \frac{1}{2!}x_2\right)\right) \\
&= 2!\left( f'(x_0)\frac{1}{2!}x_2 + \frac{1}{2}f''(x_0)x_1 x_1 \right) \\
&= f'(x_0)x_2 + f''(x_0)x_1 x_1 \\
&= f'(x_0)\frac{d^2 x}{dt^2} + f''(x_0)\left(\frac{\mathrm{d}x}{\mathrm{d}t}\right)^2 \qquad (18) \\
&= \frac{d^2}{dt^2}f(x(t)) \qquad\qquad\qquad\qquad\quad (19)
\end{aligned}
$$

$$
\begin{aligned}
y_3 = y_3(x_0, x_1, x_2, x_3) \qquad &= 3!\left( y_{[3]}\left(x_0, x_1, \frac{1}{2!}x_2, \frac{1}{3!}x_3\right)\right) \\
&= 3!\left( f'(x_0)\frac{1}{3!}x_3 + f''(x_0)x_1\frac{1}{2!}x_2 + \frac{1}{6}f'''(x_0)x_1 x_1 x_1 \right) \\
&= f'(x_0)x_3 + 3f''(x_0)x_1 x_2 + f'''(x_0)x_1 x_1 x_1 \\
&= f'(x_0)\frac{d^3 x}{dt^3} + 3f''(x_0)\frac{\mathrm{d}x}{\mathrm{d}t}\frac{d^2 x}{dt^2} + f'''(x_0)\left(\frac{\mathrm{d}x}{\mathrm{d}t}\right)^3 \qquad (20) \\
&= \frac{d^3}{dt^3}f(x(t)) \qquad\qquad\qquad\qquad\qquad\qquad\qquad\quad (21)
\end{aligned}
$$

$$\vdots$$

Therefore, eqs. (16), (17), (19) and (21) show that the derivative coefficient $y_i$ are exactly the $i$th order higher derivatives of the composition $f(x(t))$ with respect to $t$. The key insight to this exercise is that by writing the derivative coefficients explicitly we reveal that the expressions for the terms, eqs. (16) to (18) and (20), involve terms previously computed for lower order terms.

In general, it will be useful to consider that the $y_k$ derivative coefficients is a function of all lower order input derivatives

$$
y_k = y_k(x_0, \ldots, x_k). \tag{22}
$$

We provide the API to compute this in JAX by indexing the $k$-output of `jet`

$$
y_k = \mathtt{jet}(f, x_0, (x_1, \ldots, x_k))[k].
$$

## A.2 Relationship with Differential Equations

### A.2.1 Autonomous Form

We can transform the initial value problem

$$
\frac{\mathrm{d}x}{\mathrm{d}t} = f(x(t), t) \quad \text{where} \quad x(t_0) = x_0 \in \mathbb{R}^n \tag{23}
$$

into an *autonomous* dynamical system by augmenting the system to include the independent variable with trivial dynamics Hairer et al. (1993):

$$
\frac{\mathrm{d}}{\mathrm{d}t}\begin{pmatrix} x \\ t \end{pmatrix} = \begin{pmatrix} f(x(t)) \\ 1 \end{pmatrix} \quad \text{where} \quad \begin{pmatrix} x(0) \\ t(0) \end{pmatrix} = \begin{pmatrix} x_0 \\ t_0 \end{pmatrix} \in \mathbb{R}^n \tag{24}
$$

We do this for notational convenience, as well it disambiguates that derivatives with respect to $t$ are meant in the "total" sense. This is aleviates the potential ambiguity of $\frac{\partial}{\partial t} f(x(t), t)$ which could mean both the derivative with respect to the second argument and the derivative through $x(t)$ by the chain rule $\frac{\partial f}{\partial x} \frac{\partial x}{\partial t}$.

### A.2.2 Taylor Coefficients for ODE Solution with `jet`

Recall that `jet` gives us the coefficients for $y_i$ as a function of $f$ and the coefficients $x_{j \leq i}$. We can use `jet` and the relationship $x_{k+1} = y_k$ to recursively compute the coefficients of the solution polynomial.

---

**Algorithm 1** Taylor Coefficients for ODE Solution by Recursive Jet

---

```
# Have: x_0, f
# Want: x_1, ..., x_K

y_0 = jet(f, x_0, (0,)) # equivalently, f(x_0)
x_1 = y_0

for k in range(K):
  (y_0, (y_1,..., y_k)) = jet(f, x_0, (x_1,..., x_k))
  x_{k+1} = y_k

return x_0, (x_1, ..., x_K)
```

---

### A.3 Regularizing Taylor Terms

Computing the Taylor coefficients for the ODE solution as in algorithm 1 will give a local approximation to the ODE solution. If infinitely many Taylor coefficients could be computed this would give the exact solution. The order of the final Taylor coefficient, determining the truncation of the polynomial, gives the order of the approximation.

If the higher order Taylor coefficients of the solution are large, then truncation will result in a local approximation that quickly diverts from the solution. However, if the higher Taylor coefficients are small then the local approximation will remain close to the solution. This motivates our regularization method. The effect of our regularizer on the Taylor expansion of a solution to a neural ODE can be seen in fig. 9.

## Appendix B   Experimental Details

Experiments were conducted using GPU-based ODE solvers. Training gradients were computed using the adjoint method, in which the trajectory is reconstructed backwards in time to save memory, for backpropagation. As in Finlay et al. (2020), we normalize our regularization term in eq. (1) by the dimension of the vector-valued trajectory $\mathbf{z}(t)$ so that we may choose $\lambda$ free of scaling by the dimension of the problem.

### B.1 Efficient computation of the gradient of regularization term

To optimize our regularized objective, we must compute its gradient. We use the adjoint method as described in Chen et al. (2018) to differentiate through the solution to the ODE. In particular, to optimize our model we only need to compute the gradient of the regularization term. The adjoint method gives the gradient of the ODE solution as a solution to an augmented ODE.

Figure 9: *Left:* The dynamics and a trajectory of a neural ODE trained on a toy supervised learning problem. The dynamics are poorly approximated by a 6th-order local Taylor series, and requires 92 NFE by a solve by a 5th-order Runge-Kutta solver. *Right:* Regularizing the 6th-order derivatives of trajectories gives dynamics that are easier to solve numerically, requiring only 68 NFE.

## B.2 Supervised Learning

The dynamics function $f : \mathbb{R}^d \times \mathbb{R} \to \mathbb{R}^d$ is given by an MLP as follows

$$
\begin{aligned}
z_1 &= \sigma(x) \\
h_1 &= W_1[z_1; t] + b_1 \\
z_2 &= \sigma(h_1) \\
y &= W_2[z_2; t] + b_2
\end{aligned}
$$

Where $[\cdot; \cdot]$ denotes concatenation of a scalar onto a column vector. The parameters are $W_1 \in \mathbb{R}^{h \times d}, b_1 \in \mathbb{R}^h$ and $W_2 \in \mathbb{R}^{d \times h}, b_2 \in \mathbb{R}^d$. Here we use 100 hidden units, i.e. $h = 100$. We have $d = 784$, the dimension of an MNIST image.

We train with a batch size of 100 for 160 epochs. We use the standard training set of 60,000 images, and the standard test set of 10,000 images as a validation/test set. We optimize our model using SGD with momentum with $\beta = 0.9$. Our learning rate schedule is `1e-1` for the first 60 epochs, `1e-2` until epoch 100, `1e-3` until epoch 140, and `1e-4` for the final 20 epochs.

## B.3 Continuous Generative Modelling of Time-Series

The PhysioNet dataset consists of observations of 41 distinct traits over a time period of 48 hours. We remove the parameters "Age", "Gender", "Height", and "ICUType" as these attributes do not vary in time. We also quantize the measurements for each attribute by the hour by averaging multiple measurements within the same hour. This leaves 49 unique time stamps (the extra time stamp for observations at exactly the endpoint of the 48 hour observation period). We report all our losses on this quantized data. We performed this rather coarse quantization for computational reasons having to do with our particular implementation of this model. The validation split was obtained by taking a random split of 20% of the trajectories from the full dataset. In total there are 8000 trajectories. Code is included for processing the dataset, and links to downloading the data may be found in the

code for Rubanova et al. (2019). All other experimental details may be found in the main body and appendices of Rubanova et al. (2019).

### B.4 Continuous Normalizing Flows

For the model trained on the MINIBOONE tabular dataset from Papamakarios et al. (2017), we used the same architecture as in Table 4 in the appendix of Grathwohl et al. (2019). We chose the number of epochs and a learning rate schedule based on manual tuning on the validation set, in contrast to Grathwohl et al. (2019) who tuned these automatically using early stopping and an automatic heuristic for the learning rate decay using evaluation on a validation set. In particular, we trained for 500 epochs with a learning rate of `1e-3` for the first 300 epochs, `1e-4` until epoch 425, and `1e-5` for the remaining 75 epochs. The number of epochs and learning rate schedule was determined by evaluating the model on the validation set every 10 epochs, and decaying the learning rate by a factor of 10 once the loss on the validation set stopped improving for several evaluations, with the goal of matching or improving upon the log-likelihood reported in Grathwohl et al. (2019). The data was obtained as made available from Papamakarios et al. (2017), which was already processed and split into train/validation/test. In particular, the training set has 29556 examples, the validation set has 3284 examples, and the test set has 3648 examples, which consist of 43 features.

It is important to note that we implemented a single-flow model for the MNIST dataset, while the original comparison in Finlay et al. (2020) was on a multi-flow model. This accounts for discrepancy in bits/dim and NFE reported in Finlay et al. (2020).

All other experimental details are as in Grathwohl et al. (2019).

### B.5 Hardware

MNIST Supervised learning, Physionet Time-series, and MNIST FFJORD experiments were trained and evaluated on NVIDIA Tesla P100 GPU. Tabular data FFJORD experiments were evaluated on NVIDIA Tesla P100 GPU but trained on NVIDIA Tesla T4 GPU. All experiments except for MNIST FFJORD were trained with double precision for purposes of reproducibility.

## Appendix C   Additional Results

### C.1 Overfitting of NFE

Figure 10: The difference in NFE is tracked by the variance of NFE.

In fig. 10 we note that there is a striking correspondence in the variance of NFE across individual examples (in both the train set (dark red) and test set (light red)) and the absolute difference in NFE between examples in the training set and test set. This suggests that any difference in the average NFE between training examples and test examples is explained by noise in the estimate of the true average NFE. It is also interesting that speed regularization does not have a monotonic relationship with the variance of NFE, and we speculate as to how this might interact between the correspondence of NFE for a particular example and the difficulty in the model correctly classifying it.

## C.2  Trading off function evaluations with a surrogate loss

In fig. 11 and fig. 12 we confirm that our method poses a suitable tradeoff not only on the loss being optimized, but also on the potentially non-differentiable loss which we truly care about. On MNIST, we get a similar pareto curve when plotting classification error as opposed to cross-entropy loss, and similarly on the time-series modelling task we see that we get a similar pareto curve on MSE loss as compared to IWAE loss. The pareto curves are plotted for $\mathcal{R}_3$, $\mathcal{R}_2$ respectively.

Figure 11: MNIST Classification

Figure 12: Physionet Time-Series

## C.3  Wall-clock Time

We include additional tables with wall-clock time and training with fixed grid solvers in table 3 and table 4.

# Appendix D  Comparison to How to Train Your Neural ODE

The terms from Finlay et al. (2020) are

$$\|f(\mathbf{z}(t), t, \theta)\|_2^2$$

and an estimate of

$$\|\nabla_{\mathbf{z}} f(\mathbf{z}(t), t, \theta)\|_F^2$$

Table 3: Classification on MNIST

|  | Training | | Evaluation using adaptive solvers | | | | |
|---|---|---|---|---|---|---|---|
|  | Steps | Hours | Loss | NFE | $\mathcal{R}_2$ | $\mathcal{B}$ | $\mathcal{K}$ |
| No Regularization | 2 | 0.08 | .0239 | 116 | 25.9 | .231 | 7.91 |
|  | 4 | 0.13 | .0235 | 110 | 21.9 | .234 | 7.66 |
|  | 8 | 0.23 | .0236 | 110 | 21.3 | .233 | 7.62 |
|  | $\infty$ | 1.71 | .0235 | 110 | - | .233 | 7.63 |
| RNODE | 2 | 0.12 | .0238 | 110 | 18.4 | .229 | 7.07 |
| (Finlay et al., 2020) | 4 | 0.20 | .0238 | 110 | 14.6 | .230 | 6.85 |
|  | 8 | 0.37 | .0238 | 110 | 14.1 | .229 | 6.82 |
| TayNODE (ours) | 2 | 0.19 | .0234 | 104 | 3.2 | .217 | 7.12 |
|  | 4 | 0.33 | .0234 | 104 | 2.4 | .218 | 7.06 |
|  | 8 | 0.61 | .0234 | 104 | 2.4 | .219 | 7.06 |
|  | $\infty$ | 2.56 | .0234 | 104 | - | .233 | 7.63 |

Table 4: Density Estimation on Tabular Data (MINIBOONE)

| | Training | | Evaluation using adaptive solvers | | | | |
|---|---|---|---|---|---|---|---|
| | Steps | Hours | Loss | NFE | $\mathcal{R}_2$ | $\mathcal{B}$ | $\mathcal{K}$ |
| No Regularization | 4 | 0.19 | 9.78 | 185 | 17.1 | 4.10 | 1.72 |
| | 8 | 0.37 | 9.77 | 184 | 19.0 | 4.10 | 1.77 |
| | $\infty$ | 1.64 | 9.74 | 182 | - | 4.10 | 1.77 |
| RNODE | 4 | 0.19 | 9.77 | 182 | 15.9 | 4.02 | 1.65 |
| (Finlay et al., 2020) | 8 | 0.38 | 9.76 | 181 | 17.3 | 4.01 | 1.69 |
| | 16 | 0.73 | 9.77 | 189 | 17.5 | 4.03 | 1.70 |
| TayNODE (ours) | 4 | 0.49 | 9.84 | 177 | 13.1 | 4.00 | 1.57 |
| | 8 | 0.96 | 9.79 | 181 | 13.6 | 3.99 | 1.58 |
| | 16 | 1.90 | 9.77 | 181 | 13.7 | 3.99 | 1.59 |

These are combined with a weighted average and integrated along the solution trajectory.

These terms are motivated by the expansion

$$\frac{\mathrm{d}^2 \mathbf{z}(t)}{\mathrm{d}t^2} = \nabla_{\mathbf{z}} f(\mathbf{z}(t), t) f(\mathbf{z}(t), t) + \frac{\partial f}{\partial t}(\mathbf{z}(t), t)$$

Namely, eq. (3) regularizes the first total derivative of the solution, $f(\mathbf{z}(t), t)$, along the trajectory, and eq. (4) regularizes a stochastic estimate of the Frobenius norm of the spatial derivative, $\nabla_{\mathbf{z}} f(\mathbf{z}(t), t)$, along the solution trajectory.

In contrast, $\mathcal{R}_2$ regularizes the norm of the second total derivative directly. In particular, this takes into account the $\frac{\partial f}{\partial t}$ term. In other words, this accounts for the explicit dependence of $f$ on time, while eq. (3) and eq. (4) capture only the implicit dependence on time through $\mathbf{z}(t)$.

Even in the case of an autonomous system, that is, where $\frac{\partial f}{\partial t}$ is identically 0 and the dynamics $f$ only depend implicitly on time, these terms still differ. Namely, $\mathcal{R}_2$ integrates the following along the solution trajectory:

$$\|\nabla_{\mathbf{z}} f(\mathbf{z}(t), t, \theta) f(\mathbf{z}(t), t, \theta)\|_2^2$$

while Finlay et al. (2020) penalizes the respective norms of the matrix $\nabla_{\mathbf{z}} f(\mathbf{z}(t), t)$ and vector $f(\mathbf{z}(t), t)$ separately.