[Reviews · NeurIPS 2020]

Review 1

Summary and Contributions: This paper presents a regularization scheme for neural ordinary differential equations (ODE) that favors easier-to-solve dynamics. Given a k'th order ODE solver, the authors propose to augment the loss term with an integral over k'th-order state derivatives (or its squared norm). The model is tested on classification, time-series modeling, and continuous-time normalizing flow tasks, where the number of function evaluations is significantly decreased while retaining a comparable test performance. The authors also describe how to efficiently compute higher-order derivatives with Taylor mode auto differentiation (AD), and provide an open-source implementation.

Strengths: As claimed, the proposed regularization scheme leads to simpler vector fields so that integration with adaptive step ODE solvers takes a smaller amount of time. The integral needed to compute the regularization term is computed in parallel with the ODE system, which is pretty convenient. The experiments are quite extensive, all applications of neural ODEs are covered.

Weaknesses: - The contributions in this paper are twofold: (i) The new regularization term, and (ii) an efficient implementation of it. Despite being different from eq. (1), augmenting the loss with an extra penalty term is already proposed in Finlay et al (2020). Similarly, Taylor mode AD was described in [1]. As such, I doubt the paper would be a significant contribution to neural ODE literature. - As pointed out by the authors, presented method does not improve the training time, which is usually the bottleneck in generative models. In fact, looking at Table 2, second to the last RNODE row balances the train time/NFE and log likelihood the best. An additional ~19 hours of training to reduce the test NFE is not a sensible trade-off to me. Finally, not surprisingly, both RNODE and TayNODE do a better job at minimizing respective objectives (R_2 and B/K). - The results on time series modeling are not so impressive. Vector fields in Figure 4 look rather similar compared to, e.g., those in Figure 1. Also, 8% performance drop to save compute time *only* in test mode is not so preferable. [1] Bettencourt, Jesse, Matthew J. Johnson, and David Duvenaud. "Taylor-Mode Automatic Differentiation for Higher-Order Derivatives in JAX." (2019).

Correctness: The claims and the experiments go hand in hand. I did not find any flaw in the experiment setup. My only comment is that looking at Figure 6, it's indeed that setting K=2 when the integrator is second order gives the best tradeoff. But otherwise, there is no apparent tie between the order of regularization and solver, which contradicts with lines 99-100.

Clarity: The paper is well-written and organized.

Relation to Prior Work: Overall, the related work section is well-written. As mentioned before, a citation to [1] would be nice.

Reproducibility: Yes

Additional Feedback: I decided to increase my score after reading the rebuttal and discussing with other reviewers. - The authors and I are in agreement with the drawbacks of the increased training time. It's indeed that test execution time drops if this methodology is applied. Although, personally, I'm not aware of any real-time test deployment of neural ODEs where the execution time is crucial, I give credit to the authors in that. - I now better see the distinctions between this work and Finley et al. 2020. Although the starting point of Finley et al. 2020 is different from this paper (optimal transport perspective), the introduced recipe leads to simpler (possibly non-stiff) dynamics, very much like in this paper. What's more, the presented results still does not improve much upon Finley et al 2020; I'm not sure which method I'd choose to penalize a NODE vector field. I'd definitely like to see some mesaure of error uncertainty since the losses achieved by vanilla NODE, RNODE and TayNODE are very close (particularly in the tables in the appendix). - I guess I wasn't clear in what I said. Essentially, my message is that the regularized vector field in Fig.4 looks as complicated as the unregularized one in Fig.1, which demonstrates the benefits of the proposed method in time-series models is limited.


Review 2

Summary and Contributions: The paper proposes a new regularization technique in Neural ODE models that make the black-box dynamics easier to solve. A thorough empirical evaluation is presented to assess the benefits and pitfalls of this new regularizer. A heads-up: I wasn't able to open the Supplementary PDF on my computer (a Windows machine). Post-rebuttal update: I'm glad that the authors appreciate my suggestions. I'm looking forward to read the revised version. My score for now remains unchanged high.

Strengths: The paper addresses an important need when training Neural ODEs models. The paper is very critical about its own methodology and thoroughly investigates the benefits and pitfalls.

Weaknesses: Some parts of the presentation are kept short borderline of being too short to be complete. However, I hope to provide concrete constructive suggestions below.

Correctness: The theoretical contributions are presented as heuristics, but the rationale is made clear. The empirical methodology seems exhaustive and appropriate.

Clarity: Clarity is the reason for my low score. Presentation is typically highly subjective, so I do understand if the authors or the other reviewers disagree with some or all points and I do not consider any of them to be sufficient reasons for rejection. This list will be sorted chronologically, not according to severity. l. 47: throughout the intro, the authors often use qualifiers for their claims except here. Often these qualifiers are not needed (e.g., line 30), but here I think a qualifier is necessary, as error bounds and actual error can be quite different things. Paragraph at l. 64: The paragraph does not mention that the higher-order is achieved by matching the numerical and the analytical derivatives of the solution at t+h. For uninitiated readers, this might be a crucial omission. l. 77: Here, advantages of Neural ODEs are mentioned whereas throughout the rest of the text, the authors are on the defense for their general applicability. I propose to move the citations of Sect. 8 here, delete Sect. 8 and worry less about whether people question the usage of Neural ODEs in general. (Also, in the broader impact section, please remove the word "better".) l. 130: the Taylor mode AD makes or breaks the regularization technique in practice at the moment, yet only half a paragraph and a book chapter as citation are given. Me and probably many readers would like to learn more about Taylor mode AD here. Also, I probably should have worked out some higher-order derivatives in my head to understand Table 1, but in the limited time, the examples didn't really improve my understanding. Fig. 4: as you stated earlier, there are degrees of freedom to be expected in the vector field, so plotting two examples next to each other without any performance plots is leaving much to the imagination. For example, in (b), the sharp turn in the area z_1 approx 0, z_0 < 0 looks like it might induce stiffness issues. Sect. 5 and later: the submission seems to awkwardly justify the higher computation time, in particular since NFEs are so strongly emphasized in the beginning. I would suggest to be more explicit once in the beginning and then not to worry about it for the rest of the manuscript too much. Sect. 6.4: I think I was able to understand this bit plus Fig. 8 (a), but this is probably a bit too dense. How are the y-values computed in Fig. 8 (a)?

Relation to Prior Work: I think a connection to https://openreview.net/forum?id=B1e9Y2NYvS should be made, otherwise it looks okay.

Reproducibility: Yes

Additional Feedback: Do you think that your heuristic might also affect the robustness of Neural ODEs? Cf. https://openreview.net/forum?id=B1e9Y2NYvS and https://arxiv.org/abs/2006.05749 Fig. 5 (b): it looks like there is a large stochastic effect based on the scatter. Maybe the authors should comment how sensitive all the experiments are on stochasticity overall (from the rest of the plots I get the impression that effects are mostly robust) Fig. 6 (d): did you make an analysis what the distribution of chosen orders have been in the adaptive order method?


Review 3

Summary and Contributions: This paper proposes a new training method for neural ODEs to improve the test performance by reducing the number of function evaluations (NFE) by adding the norm of higher order derivative as a penalty term to the original objective function.

Strengths: The idea of training a neural ODE so that a solver can easily solve it is novel and interesting. The goal of reducing the computational cost without losing solution quality is very practical.

Weaknesses: The proposed method does not seem mature enough. The merits of the proposed regularization are not clear in the experiments, and not convincing enough.

Correctness: Maybe yes.

Clarity: This paper could be written better. There are unnecessarily many subsections.

Relation to Prior Work: Yes. The difference from other works is relatively clear.

Reproducibility: Yes

Additional Feedback: The idea of learning neural ODE that can be solved easily is interesting. While this paper focuses on reducing the test time, I wonder if there are some other advantages of suppressing higher-order derivatives for some applications.


Review 4

Summary and Contributions: This paper considers the problem of fitting an ODE to data when the vector field is parametrised by a neural network. The main contribution is to regularise the data fitting criterion in a manner which biases the estimate towards a vector field that makes the underlying ODE well-behaved.

Strengths: The authors discuss the trade-off between many important quantities such as the number vector field evaluations, degree of regularisation, unregularised loss, and computation time at training versus test time.

Weaknesses: The relation between the present approach and prior literature on function approximation via Gaussian processes / Kernel interpolation is left completely explored. I elaborate on this in the comments below.

Correctness: Appears so.

Clarity: Well enough.

Relation to Prior Work: Yes.

Reproducibility: Yes

Additional Feedback: Some comments: 1) The regulariser R_K is precisely the square RKHS norm for (K-1)-times integrated Wiener processes on [t_0,t_1 starting at zero [Sec 10, 1]. From this perspective the regulariser enforces the resulting ODE solution to have a small norm in this RKHS. However, pinning the Wiener process is known to give poor approximation properties of smooth functions wherefore a "released version" is often preferred [Sec 10, 1]. I think the paper would benefit from a discussion on whether this is applicable to the present approach as well. 2) R_K is also the Onsager--Machlup functional = "prior density" functional associated with (K-1)-times integrated Wiener processes starting at zero [2]. Consequently it seems like the regularised objective in Eq. (2) could have an interpretation of some kind of constrained maximum a posteriori estimate. Is this the case? 3) In Sec 6.3 it is mentioned that while evaluation at test time is generally faster with the present approach, the training time is in general slower. I think this could have been explored in more detail. That is, how many tests per train do I need to perform for the present approach to offer an advantage? [1] A. W. van der Vaart and J. H. van Zanten "Reproducing kernel Hilbert spaces Gaussian priors" (2008). [2] D. A. Dutra, B. O. S. Teixeira, and L. A. Aguirre "Maximum a posteriori state path estimation: Discretization limits and their interpretation".

[Author Response · NeurIPS 2020]

**Reviewer 1** We thank the reviewer for their helpful comments. We agree that the increased training time is a notable disadvantage to our method. However, improving test time evaluation at the expense of increased train-time is a reasonable trade-off. In applications where the model is evaluated many more times than it is optimized the increased training cost can be justified. In response to this concern we will connect our motivation for improved test time performance by referencing the existing literature on efficient inference methods, e.g. quantization.

The reviewer's primary concern is this work's relationship with existing literature. We agree that the relationship to the workshop paper Bettencourt et al. 2019 should be made clear.

Our paper includes a detailed comparison to Finlay et al. 2020 that addresses the reviewer's concerns about our contributions. Our regularizer, like theirs, is augmenting the loss with an extra penalty term. Their regularization terms are motivated by optimal transport and reusing computation specifically in FFJORD. We show in the appendix how their regularizer can be generalized to ours, and how ours tracks the expense of the chosen adaptive solver. Our motivation can be extended to optimize properties of other solvers, e.g. stiffness. Due to these distinctions, our method is a contribution to the literature. Empirically, we extensively compare to Finlay et al. 2020 in our experiments on density estimation. We also include experiments on additional domains and phenomena such as regularization effect on overfitting.

The reviewer's comment on time series results reference the similarities between Figures 1 and 4. This suggests to us that we can improve our presentation in those figures, as they are not comparable. Figure 1 is a 1D state vs time plot, whereas Figure 4 is a phase plot, and time is denoted implicitly by the arrows along the curves. Also, the figures describe different tasks with different properties. Figure 1 describes a simple output classified from the input, so the intermediate dynamics are not explicitly constrained by data and could be regularized to straight trajectories. Figure 4 describes a time series task, so the trajectories must model the data along intermediate values and are constrained from becoming straight trajectories. The result is that we can improve evaluation cost without changing the appearance of the trajectories.

As we explain in Section 6.2, we agree that $K = 3$ with a 3rd order solver shows a marginal improvement over other solvers, and there is no clear winner for order 5 (6c) or adaptive (6d). We apologize for the misleading claim on lines 99-100, and will change the wording.

**Reviewer 2** We will remove the qualifiers on l. 30 and add more detail on l. 47. We will include the detail on l. 64. Re. l. 77, we fear that we may mislead the reader about the applicability of our method without being clear about its drawbacks. We will remove "better" from Broader Impact. There is a detailed appendix section for Taylor-mode AD.

We agree that the dynamics seem stiff in Figure 4. We note that the caption of this figure summarizes the performance of the model and the improvement in NFE. Your suggestions for improving clarity in Section 5 are very appreciated. We will reorganize the discussion of NFE vs. computation time to avoid repetition. Thank you for highlighting the connection to adversarial robustness. Although we did not investigate it in this work, we think this is an interesting avenue for potential future research, and will cite these works in the main text. The noise in 5b) is from the optimization (noise which is present without our regularization). Any tuning of optimization or other hyperparameters were done on unregularized models and left unchanged when training with regularization. Unfortunately we were not able to analyze the distribution of orders chosen by the adaptive solver due to the engineering required. We thank the reviewer for raising this as we think it's an interesting question. The formula for the $y$-axis in Figure 8a) is $\log \max_i \{x_i - x_i^{\text{true}}\}$, where $x_i^{\text{true}}$ is the $i$th component of the (fixed) true solution computed using a tight tolerance for the solver, and $x_i$ is the $i$th component computed with `atol` and `rtol` parameters passed to the solver as given on the horizontal axis.

**Reviewer 4** We thank the reviewer for pointing out interesting literature we were not previously aware of. The high-level conceptual connection is a good one, but the methods and motivations are quite distinct: 1) we provide soft constraints in the form of regularization instead of a hard constraint forcing the derivatives to be exactly zero; 2) we only regularize one higher derivative, and not all of them simultaneously, so the lower derivatives need not be small even if the higher ones are; 3) in many cases the ODE trajectories need not smoothly approximate a function along the whole interval, but rather only at the endpoints, e.g. as in Figure 1 (see second last paragraph in comments for Reviewer 1).

Nevertheless it is interesting to motivate our regularizer by connecting it to the notion of certain priors on paths, and we thank the reviewer for making us aware of this literature. We will cite this work in the main text. Potentially related is Figure 8. c) where we investigated the potential effect of statistical regularization from our method, and found there was in fact little effect.

As for the comment about test time, see the comments for Reviewer 1.

[Meta-Review · NeurIPS 2020]

The paper introduces a regularization scheme for neural ODE based on state derivatives of the ODE that favor simpler parameterized vector fields and then allow the control of complexity/ performance of Neural ODE solvers at test time. The reviewers agree that this is an additional step towards the deployment of Neural ODE models and that the specific topic of integration time control has not been much explored for now. The experimental analysis is extensive and explores different merits/ pitfalls of the model. The different issues mentioned in the reviews gave rise to extensive discussions among the reviewers that helped clarify most of the issues. Overall this is considered as a positive and new contribution to the topic. The authors are encouraged to further develop the motivations for the regularization terms that remain largely heuristic in the paper and to clarify the presentation according to the remarks.